# Effect of Face Masks on Physiological and Perceptual Responses during 30 Minutes of Self-Paced Exercise in Older Community Dwelling Adults

**DOI:** 10.3390/ijerph191912877

**Published:** 2022-10-08

**Authors:** Grace Vogt, Kimberley Radtke, Andrew Jagim, Dominique Peckumn, Teresa Lee, Richard Mikat, Carl Foster

**Affiliations:** 1Department of Exercise and Sport Science, University of Wisconsin-La Crosse, La Crosse, WI 54601, USA; 2Sports Medicine, Mayo Clinic Health System, Onalaska, WI 54650, USA; 3Department of Health, Exercise & Rehabilitative Sciences, Winona State University, Winona, MN 55987, USA

**Keywords:** surgical mask, N95-mask, middle-older aged individuals, submaximal exercise, performance measures

## Abstract

This study examined the effects of different types of masks (no mask, surgical mask (SM), and N95-mask) on physiological and perceptual responses during 30-min of self-paced cycle ergometer exercise. This study was a prospective randomly assigned experimental design. Outcomes included workload (Watts), oxygen saturation (SpO_2_), end-tidal carbon dioxide (PetCO_2_), heart rate (HR), respiratory rate (RR), rating of perceived exertion (RPE), and rating of perceived dyspnea (RPD). Volunteers (54–83 years (*n* = 19)) completed two familiarization sessions and three testing sessions on an air braked cycle ergometer. No significant difference was found for condition x time for any of the dependent variables. RPE, RPD, and PetCO_2_ were significantly higher with an N95-mask vs. no mask (NM) ((*p* = 0.012), (*p* = 0.002), (*p* < 0.001)). HR was significantly higher with the SM compared to the NM condition (*p* = 0.027) (NM 107.18 ± 9.96) (SM 112.34 ± 10.28), but no significant difference was found when comparing the SM to the N95 condition or when comparing the N95condition to the NM condition. Watts increased across time in each condition (*p* = 0.003). Initially RR increased during the first 3 min of exercise (*p* < 0.001) with an overall gradual increase noted across time regardless of mask condition (*p* < 0.001). SpO_2_ significantly decreased across time but remained within normal limits (>95%). No significant difference was found in Watts, RR, or SpO_2_ regardless of mask condition. Overall, the N95mask was associated with increased RPE, RPD, and PetCO_2_ levels. This suggests trapping of CO_2_ inside the mask leading to increased RPE and RPD.

## 1. Introduction

Since the start of the COVID-19 outbreak, the implications of mask wearing have been of interest for researchers as government mandated shutdowns, mask mandates, and resistance to mask use have occurred. However, facemasks serve as an important preventative measure against COVID-19 transmission [1,2]. Facemasks have traditionally been used in countries with high population densities and poor air quality, but they have become more important recently for use by some countries as a preventive measure against infectious diseases [3].

In response to the COVID-19 pandemic, as well as to potential future infectious disease outbreaks, it is important to understand how mask usage influences not only safety and disease prevention but also quality of life. One such instance is how mask wearing may influence exercise capacity, specifically ventilation, VO_2_ kinetics, and exercise tolerance. During the peak of the COVID-19 pandemic, facemasks were recommended in group settings during exercise, particularly while exercising indoors, such as in fitness centers. When completing a bout of exercise, recent research found that face mask wearing had no clinically significant impact on total distance traveled or time to completion compared to exercising without a face mask across a variety of populations [4,5,6,7,8,9,10,11,12,13]. However, in contrast, two studies have found decreased workloads while wearing a mask [14,15]. For example, Driver et al. found that cloth face masks reduced exercise time by 14% and VO_2max_ by 29% and negatively impacting oxygen saturation (SpO_2_) and heart rate (HR) [15]. While HR and respiratory rate (RR) typically increase across time during exercise, contradictory results have been identified with various masked conditions, indicating that mask wearing may influence HR and RR responses during various exercise conditions [4,5,6,7,15,16,17].

It is possible that different mask types may exert varying influences on the physiological effects of wearing a mask during physical activity. A primary concern for those exercising while wearing a mask is that some investigators have reported that an N95 mask may increase inhaled carbon dioxide concentration (F_i_CO_2_), reduce inspired oxygen concentration (F_i_O_2_), and increase the work of breathing [13,18,19,20,21,22]. Increased F_i_CO_2_ can contribute to fatigue, dizziness, shortness of breath, headache, and other negative cardiorespiratory or nervous system changes in the body [23,24]. There has also been evidence of increased respiratory resistance with N95 mask usage [14,17] as Lee and Wang found that N95 masks increased respiratory resistance by over 100% within 30-s [25]. These authors suggested that N95 masks are associated with the development of acute and/or intermittent hypoxia and hypercapnia, which may lead to an increased risk of arrhythmias during exercise [26].

Other concerns regarding mask wearing during exercise are an increase in perceived levels of heat, humidity, and discomfort that may discourage mask usage. While surgical masks (SM) allow for greater breathability, the more clinically effective N95 masks often have a tighter face seal, which has been associated with higher perceived dyspnea scores [4,5,8,9,11]. Furthermore, N95 masks have been shown to increase facial skin temperature, heat, humidity, and breathing difficulty more than other types of masks. In turn, these alterations may lead to greater discomfort compared to wearing no mask (NM) or a SM [12,14,27,28]. The perceived breathing discomfort can lead to increased ratings of perceived exertion (RPE) while exercising. Overall, RPE has been found to be significantly higher with masks vs. no mask [5,15,29]. It is also important to note that the duration of mask usage seems to play a role in perceived discomfort as the longer the duration of wear, the higher the level of discomfort [30,31].

While the effects of mask wearing during acute bouts of aerobic exercise in young-adult populations has been evaluated, research examining the effects of mask wearing during longer-duration self-paced exercise performance and the subsequent physiological outcomes is lacking, particularly in older adults for whom mask wearing may be more clinically relevant. This is especially true for older adult populations who may have reductions in baseline exercise capacity but have a higher need for mask usage secondary to the much larger negative consequences of COVID-19 infection. Therefore, the purpose of the current study was to examine the effects of different types of masks on physiological and perceptual responses and tolerance when performing 30-min of self-paced exercise on a cycle ergometer. Primary outcomes included exercise workload, oxygen saturation (SpO_2_), end-tidal carbon dioxide (PetCO_2_), HR, RR, RPE, and RPD.

## 2. Materials and Methods

### 2.1. Study Design

Using a randomized, counter-balanced, cross-over design, participants completed three experimental sessions, during which time participants were asked to wear: (1) no mask; (2) a surgical mask; or (3) a N95-mask during the exercise trial. Participants were screened for COVID-19 and completed a 30-min bout of exercise on an air-braked cycle ergometer. To eliminate order effects, the order of mask conditions was counterbalanced with at least 24 h between tests. Participants reported to the laboratory at the same time of day for each session (±1 h) and were instructed to take medications as prescribed.

Each participant attended two familiarization sessions as well as three separate testing sessions. During the second familiarization session and all experimental testing sessions, data were collected at rest, as well as at 3-min increments throughout the 30-min exercise bout and at completion of the exercise session. Oxygen saturation (SpO_2_), end-tidal CO_2_ (PetCO_2_), power output (in Watts), HR, RR, RPE, and RPD were recorded throughout the exercise bout.

### 2.2. Participants

Community dwelling- adults between the ages of 54 and 83 years (*n* = 19) were recruited for this study. Sample size estimation using Cohen’s techniques [32] indicated that a minimum of 18 participants would be needed to detect a 5% difference in power output with a power of 0.80 and an alpha level of 0.05. Participants were eligible to participate in the study if they were between the ages of 50–90 years of age and could tolerate exercising for 30-min without experiencing chest pain/pressure, excessive shortness of breath, or pain in the back, legs, or neck. Participants were excluded from the study if they experienced unstable angina or myocardial infarction in the past month, had a history of chronic obstructive pulmonary disease (COPD), a resting HR greater than 120 beats per minute, a resting systolic blood pressure (BP) greater than 180 mm Hg, a resting diastolic BP greater than 100 mm Hg, current problems with dizziness, fainting or blackouts, were current smokers, or had an active COVID-19 infection.

Explanation of the testing protocol was verbally reviewed, and each participant was provided the opportunity to ask questions. Participants provided written informed consent and approval of the protocol was obtained from the Institutional Review Board for the Protection of Human Participants at the University of Wisconsin- La Crosse (IRB approval #21-KR-255).

### 2.3. Familiarization Session

During the first familiarization session, each participant completed a COVID-19 screening questionnaire, Physical Activity Readiness Questionnaire (PAR-Q+) [33], ACSM Exercise Pre-Participation Health Screening Questionnaire [34], physical activity questionnaire (NASA PA-R) [35] and answered subjective questions relating to perceptions on mask usage. Height, body mass, and blood pressure were measured followed by a completion of a session to predict ventilatory threshold (VT). This session involved pedaling on an air-braked cycle ergometer for a stepwise incremental test starting at a power output of 15 Watts, with an increase of 15 Watts at the end of each 2-min stage. Participants recited, outload, “The Rainbow Passage”, a 102-word speech-provoking stimulus [36,37], and also reported RPE (Borg 6–20) during the last 30-sec of each 2-min stage. Predicted VT was identified when participants were in the first Equivocal stage when reciting “The Rainbow Passage” using previously validated methods [38,39,40]. Power output at predicted VT was used as the recommended starting exercise intensity for the experimental testing sessions. During familiarization session two, participants performed a pulmonary function test (PFT) followed by a 30-min bout of self-paced exercise on the air-braked cycle ergometer to account for the learning effect [41].

VO_2_ prediction of ventilatory threshold (VT) was calculated from ACSM metabolic equations [34]. Estimated VO_2max_ was calculated using RPE collected from the predicted VT session [38,42] and extrapolated to RPE = 19 using individual linear regression to estimate VO_2max_ [39,43]. The NASA PAR-R questionnaire was also used to estimate general physical activity levels in the past month and to estimate VO_2max_ without the use of maximal testing [35].

### 2.4. Procedures

#### 2.4.1. Mask Type

The SM was a three-layer disposable surgical face mask and the N95 mask was a Kimtech^TM^ N95 Pouch Respirator. Participants followed proper fitting instructions according to each mask to ensure a tight face seal was obtained.

#### 2.4.2. Exercise Bout

During each testing session, participants were given a metronome driven cadence to pedal at according to their predicted VT for the first minute of exercise. After the first minute, the metronome was turned off and participants were instructed to self-pace for the remainder of the session. Participants were instructed to pedal at a comfortable rate, as if they were exercising for fitness, and to avoid fatigue and breathlessness. Participants did not utilize the arm handles on the ergometer to minimize upper body assistance while performing the exercise session. A chair was placed on each side of the participant while biking for stability.

#### 2.4.3. Pulmonary Function

Participants completed PFT testing (Parvo Medics TrueOne 2400) to measure lung volumes and capacities. PFTs were conducted using spirometry in which participants were asked to maximally inhale followed by a forceful exhalation as fast and as long as possible. Participants were allowed three practice trials before completing three testing trials. The best effort was used for analysis. Pulmonary function tests were evaluated according to %predicted forced expiratory volume in one second (FEV_1_), forced vital capacity (FVC), and FEV_1_/FVC using normative reference values [44].

#### 2.4.4. Heart Rate, Oxygen Saturation and End-Tidal CO_2_

Heart rate, SpO_2_ and PetCO_2_ were collected via the Phillips Heartsmart MRx. A nasal cannula (Microstream Smart CapnoLine Plus O_2_) was inserted into the participants nostrils according to product specifications. During the mask conditions, the face mask was fitted onto the participant over the nasal cannula while still forming a tight face seal with the mask according to face mask fitting instructions.

#### 2.4.5. Rating of Perceived Exertion and Dyspnea

Participants were instructed using standardized verbiage how to read and utilize the RPE (Borg RPE scale) and RPD (Modified Borg Scale) scales. Participants were asked to give RPE and RPD levels every 3 min during each exercise session. Session RPE and session RPD were also collected at the end of each bout of exercise [45].

#### 2.4.6. Subjective Questionnaire

Subjective questions asked prior to starting the testing sessions included, “Q1. Do you think a mask will impair your work?”, “Q2. Do you think wearing a mask will elevate your HR rate or RR during exercise?”, “Q3. Do you think wearing a mask will make it harder to exercise?”, and “Q4. If given a choice, would you prefer to wear a mask?”.

#### 2.4.7. Power Output

Power output was reported as Watts. This metric was recorded from the monitor screen display on the air-braked cycle ergometer.

### 2.5. Statistics

Statistical analyses were performed using SPSS Statistics for Windows, Version 28.0. Armonk, NY: IBM Corp). All data were screened for accuracy, completion, and normality. A repeated measures two-way analysis of variance (ANOVA) (condition [3] x time [11]) was used to analyze variables across time (Watts, HR, RR, PetCO_2_, RPE, RPD, and SpO_2_). If there was a significant *F* ratio for a condition x time interaction, a Bonferroni correction was conducted to assess pairwise comparisons. A Kruskal –Wallis one-way ANOVA was used to analyze overall session RPE and session RPD scores between groups. Results are presented as mean ± SD, and the significance level was set for alpha at *p* < 0.05.

## 3. Results

A total of 19 participants (9 males, 10 females) completed the study. Participant descriptive characteristics are presented in Table 1. Participants ranged from 54 to 83 years of age. PAR-Q+ and ACSM pre participation screenings identified four participants with a history of cardiovascular disease, six took prescribed medication for hypertension, four for hyperlipidemia, two participants had a history of sleep apnea, two with multiple sclerosis, one with Type 1 diabetes, one with type 2 diabetes, and two with vertigo.

Pulmonary function testing indicated all participants were within normal range of their predicted values for FEV_1_/FVC. However, FVC values in three male participants were < 80% of predicted and two of the same males also had FEV_1_ values < 80% of predicted [46]. PFT results from all participants were included in data analysis. Physical activity levels (0–7) averaged 3.2 ± 1.6 for men and 3.0 ± 1.4 for women [35].

Responses to subjective questions were categorized into “No”, “Maybe” and “Yes”. Responses to Q1-Q4 are as follows. Q1: 10 (55.5%); “No”, 1 (0.06%); “Maybe”, 7 (38.9%); “Yes”; Q2: 11 (61.1%); “No”, 0 (0.00%); “Maybe”, 7 (38.9%); “Yes”; Q3: 5 (27.8%); “No”, 2 (11.1%); “Maybe”, 11 (61.1%); “Yes”; Q4: 18 (100.0%) “No”, 0 (0%); “Maybe”, 0 (0%); “Yes”. Power output (Watts) is presented in Figure 1. No significant difference was found in workload performed between masked conditions (NM v. SM *p* = 0.978, NM v. N95 *p* = 0.103, SM v. N95 *p* = 0.133). Overall power output (Watts) significantly increased across time (*p* = 0.003). No significant difference was found between condition x time with any of the dependent variables. 

A summary of HR and RR changes across time are presented in Figure 2. No significant condition x time effect was observed for HR or RR. HR significantly increased across time in all conditions (*p* < 0.001). HR was significantly higher with the SM compared to the NM condition (*p* = 0.027) (NM 107.18 ± 9.96) (SM 112.34 ± 10.28), but no significant difference was found when comparing the SM to the N95 condition (*p* = 0.368) or when comparing the N95 mask condition to the NM condition (*p* = 0.080). RR was significantly increased during the first 3 min of exercise (*p* < 0.001) with a continued gradual increase across time regardless of mask condition (*p* < 0.001). No significant difference was found in RR between masked conditions (NM v. SM *p* = 0.430, NM v. N95 *p* = 0.538, SM v. N95 *p* = 0.456).

A summary of SpO_2_ and PetCO_2_ changes across time for each condition are found in Figure 3. No significant condition x time effect was observed for SpO_2_ or PetCO_2_. No significant difference was found in SpO_2_ between masked conditions (*p* = 0.05). SpO_2_ significantly decreased across time (*p* < 0.001) but remained within normal limits (> 95%). PetCO_2_ significantly increased from rest during the first 3 min of exercise (*p* < 0.001) and plateaued for the duration of the exercise session. PetCO_2_ was significantly higher in the N95 Mask and SM condition compared to the NM condition (*p* < 0.001), significantly higher in the N95 mask compared to the SM condition (*p* < 0.001), and significantly higher in the SM condition compared to the NM condition (*p* < 0.001).

A summary of RPE and RPD values for each condition are displayed in Figure 4. No significant condition x time effect was observed for RPE or RPD. Rating of Perceived Exertion significantly increased across time in all conditions (*p* < 0.001) and was significantly higher in the N95 mask compared to the NM condition (*p* = 0.012). Session RPE (sRPE) was not different between the mask conditions (*p* = 0.707). Rating of Perceived Dyspnea significantly increased across time in all conditions (*p* < 0.001) and was significantly higher in the N95 mask condition when compared to the NM condition (*p* = 0.002) and significantly higher in the N95 mask condition vs. the SM condition (*p* = 0.027). Session RPD (sRPD) was not different across mask conditions (*p* = 0.166).

## 4. Discussion

The primary aim of the current study was to examine the physiological and perceptual effects of wearing a mask during a 30-min bout of self-paced exercise. The main findings from the current study indicate that HR was significantly higher during the SM condition compared to NM, but there no differences in HR between the N95 mask and NM condition. Furthermore, RR, and SpO_2_ levels were not different across conditions, regardless of mask type. Some of the results from this study are in alignment with previous findings. Radtke et al. failed to observe differences in HR, RR, or SpO_2_ levels regardless of masked condition in apparently healthy OCA when completing the 6MWT [4]. Similarly, Ringham et al. found similar results in college aged students while performing a 3200-m run [5]. Contrarily, a study by Kyung et al. examined 6MWT in individuals with COPD and found HR, RR and CO_2_ concentrations were higher and SpO_2_ levels were lower when wearing an N95 mask compared to a no mask condition [17]. The results of these studies suggest that mask usage does not significantly affect RR or SpO_2_ levels in healthy individuals, but people with pulmonary disorders should be cautious wearing a mask during physical activity (as it may exacerbate ventilatory symptoms).

In the current study, RPE was found to be higher during the SM and N95 mask conditions compared to no mask. These results are in agreement with Poon et al., who found individuals had higher RPE values during an incremental treadmill test while wearing a SM vs. no mask [29]. Further, Ringham et al. and Driver et al. also reported higher RPE values during mask conditions compared to no mask [5,15]. Conversely, Shaw et al. and Radtke et al. did not observe any differences in RPE with mask usage [4,7]. Collectively, these findings suggests that RPE may increase during vigorous or prolonged bouts of exercise while wearing a facemask but appear to be less impacted during shorter durations of exercise or at lighter intensities.

While similar to RPE, dyspnea is defined as experiencing labored breathing or an unpleasant feeling during inspiration and expiration. The onset of light dyspnea is common during increased levels of exercise intensities. According to Datta et al., there are three key predictors of exertional dyspnea, including work of breathing, hypoxia, and hypercarbia [47]. While the current study did not evaluate the cost of breathing directly, factors relating to hypoxia (SpO_2_) and hypercarbia (PetCO_2_) were assessed. Although SpO_2_ was not significantly different between mask conditions, there was an overall decrease throughout the duration of the exercise session, while remaining within normal limits. This trend is similar to that which was observed by Beder et al. who found a slight decrease in SpO_2_ in surgeons wearing SMs during surgeries 1–4 h in duration [48].

Mask usage may have the potential to increase CO_2_ levels due to “re-breathing” expired air during physical activity. Increases in PetCO_2_ can be associated with either a decrease in CO_2_ removal or increases in production. Normal PetCO_2_ values range between 35 and 45 mmHg and tend to increase during exercise levels at intensities below VT [49]. The results from the current study demonstrate an increase in PetCO_2_ while wearing both a SM and N95 mask. These results are similar to those by both Epstein et al. and Rudi et al. who found exercising with an N95 mask was associated with an increase in the partial pressure of CO_2_ (PetCO_2_) levels inside the mask during incremental performance on a cycle ergometer [13,22]. Similarly, Roberge et al. found a rise in transcutaneous carbon dioxide (PtcCO_2_) levels when evaluating mask usage among healthcare workers walking for 1-hr at 1.7 mph and 2.5 mph [50]. A rise in PetCO_2_ during exercise could be explained by a trapping of CO_2_ inside the SM and N95 mask, which is associated with the individual retaining, or not being able to remove, as much CO_2_ while exercising with a mask on. This CO_2_ trapping has been linked to increased perceived dyspnea and exertion during exercise.

A secondary aim of the current study was to assess the effects of wearing a mask on work capacity. The results of our study found that power output (Watts) significantly increased over the 30-min exercise session, but no significant difference was found between mask conditions. This is similar to the results by Radtke et al. who found mask usage did not affect total distance walked when performing the 6MWT in older adults [4]. Similarly, a recent study by Just et al. examined performance measures in a 6MWT and found that wearing a surgical mask (SM) did not affect 6MWT distance of patients with advanced lung disease [51].

A subjective questionnaire was included in this study to identify participant’s pre-treatment expectations towards mask usage. About 60% of participants expected masks to make exercise feel harder, with 30–40% of participants thinking it would hinder or impact performance measures. Based on the outcomes, pre-perceptions on mask usage did not appear to significantly impact work capacity. To our knowledge, no other studies have evaluated mask usage on prolonged exercise in older adults.

### Limitations and Directions for Further Research

A few limitations were identified. The results from our study may not be generalizable to older adults with a BMI less than 25 as the average BMI for women and men from our study were classified as overweight and obese.

Tidal volume (TV) was not measured during the exercise sessions. Increasing TV is a compensatory mechanism used to increase ventilation and lower CO_2_. Knowing the tidal volume during exercise can be important to see how the individual might be compensating for increased PetCO_2_ during masked conditions. Future research should assess TV and PetCO_2_ to further explain these physiologic changes.

In addition, we did not require our participants to perform a maximal exercise bout. Although the crossover design protects against any order of effects, the number of indirect calculations used to estimate VT and VO_2_ may produce values that are slightly different from the gold standard measurements. Anormal CO_2_ response to incremental exercise is a decrease at intensities greater than VT. More research is warranted evaluating prolonged exercise, particularly above VT, in older community dwelling adults, young athletes, smokers, post-surgical patients, and pulmonary patients. However, prolonged exercise at an intensity exceeding VT in non-athletic populations is unlikely.

## 5. Conclusions

Limited research has been performed during prolonged exercise with and without a SM or a N95-mask, particularly among older adults. In summary, our study found that RPE, RPD, and PetCO_2_ were significantly higher while wearing an N95-mask compared to no mask. However, no significant difference was found in workload, RR, or SpO_2_ regardless of mask condition. The N95mask has been found to produce higher heat and humidity levels inside the mask, increased breathing resistance, and has a tighter face seal that was associated with increased RPE, RPD, and CO_2_ levels. This suggests that CO_2_ trapping inside the mask. Therefore, wearing an N95 mask may make it less comfortable for older individuals performing prolonged exercise bouts. The information from this study will be important to exercise scientists and the general public seeking to understand how SM and N95 masks effect perceptual responses and overall performance during prolonged exercise.

## Figures and Tables

**Figure 1 ijerph-19-12877-f001:**
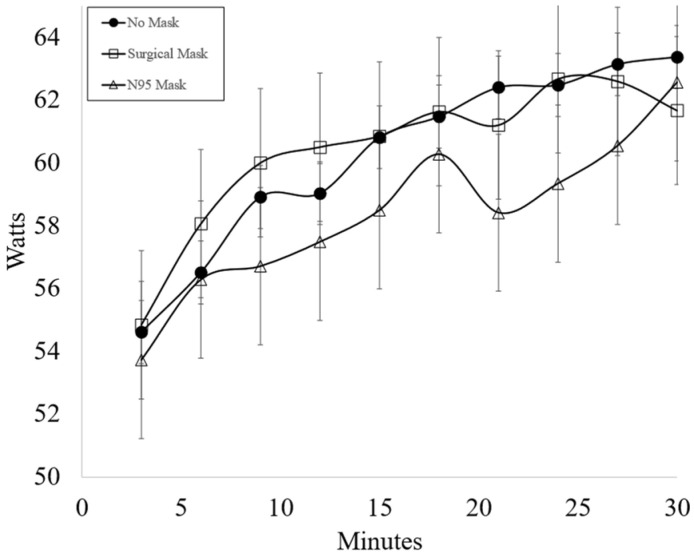
Average power output (mean ± SD) for each testing condition (*n* = 19).

**Figure 2 ijerph-19-12877-f002:**
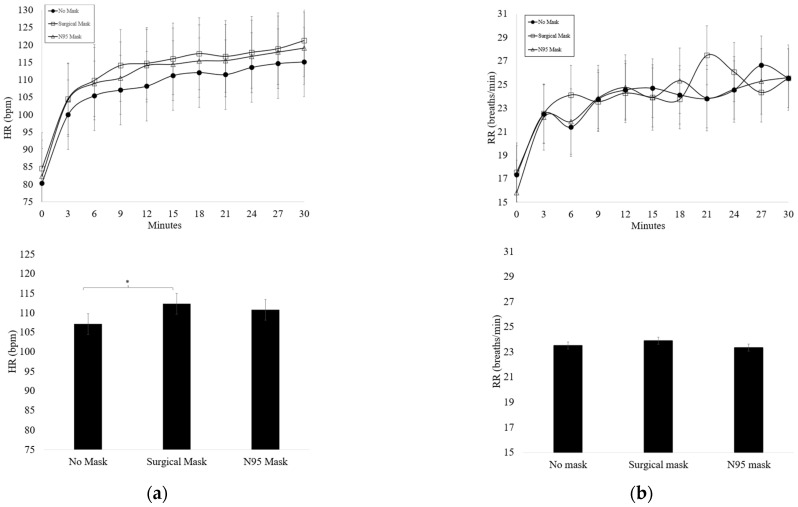
(**a**) Average heart rate (HR) (mean ± SD) during each testing condition (*n* = 19) * Significant main effect for condition (*p* < 0.05); (**b**) average respiratory rate (RR) (mean ± SD) during each testing condition (*n* = 19).

**Figure 3 ijerph-19-12877-f003:**
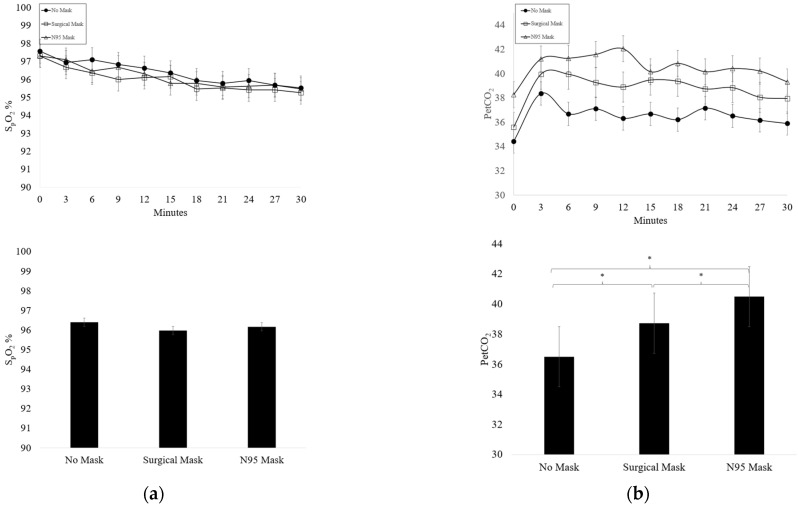
(**a**) Average oxygen saturation (SpO_2_) (mean ± SD) during each testing condition (*n* = 19); (**b**) average patient end-tidal carbon dioxide (PetCO_2_) (mean ± SD) during each testing condition (*n* = 19) * Significant main effect for condition (*p* < 0.05).

**Figure 4 ijerph-19-12877-f004:**
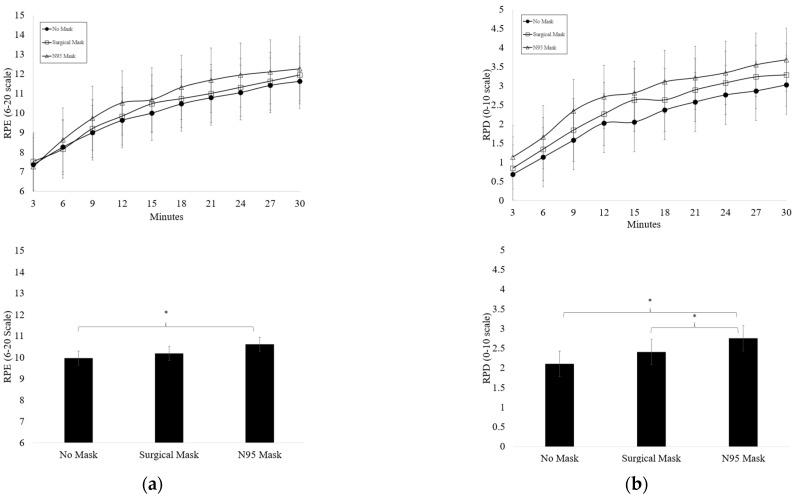
(**a**) Average rating of perceived exertion (RPE) (mean ± SD) during each testing condition (*n* = 19) * Significant main effect for condition (*p* < *0*.05); (**b**) average rating of perceived dyspnea (RPD) (mean ± SD) during each testing condition (*n* = 19) * Significant main effect for condition (*p* < 0.05).

**Table 1 ijerph-19-12877-t001:** Descriptive characteristics.

	Women (*n* = 10)	Men (*n* = 9)	Total Group (*n* = 19)
Age (years)	67.7 ± 9.2	72 ± 7.6	69.7 ± 8.5
Height (cm)	165.2 ± 4.2	175.8 ± 7.5	170.2 ± 8.0
Body Mass (kg)	73.6 ± 13.8	94.2 ± 28.9	83.4 ± 24.0
BMI (kg/m^2^)	27.0 ± 5.4	30.5 ± 8.6	28.6 ± 7.1
FEV_1_ (%)	101.7 ± 9.0	91 ± 15.3	96.6 ± 13.2
FVC (%)	99.9 ± 8.6	89 ± 15.3	94.7 ± 13.1
FEV_1_/FVC (%)	102.5 ± 5.7	100.6 ± 4.7	101.6 ± 5.2
PredictedVO_2_@VT (ml.min-1.kg-1)	16.4 ± 2.4	15.8 ± 3.4	16.1 ± 2.8
PredictedVO_2max_ (ml.min-1.kg-1)	22.8 ± 4.6	22.2 ± 6.6	22.5 ± 5.5

Data are presented as mean ± SD. cm, centimeters; kg, kilograms; m, meters; BMI, body mass index; FEV_1_, forced expiratory volume in 1 s; FVC, forced vital capacity; VO_2_, oxygen consumption; VT, ventilatory threshold.

## Data Availability

Not applicable.

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
