# Peer review of "Effect of Face Masks on Physiological and Perceptual Responses during 30 Minutes of Self-Paced Exercise in Older Community Dwelling Adults"

_ijerph, 2022, doi:10.3390/ijerph191912877_

Round 1
Reviewer 1 Report
The study is well conceived, communicated and reported. The introduction is logically developed and provides an appropriate rationale for the study. The methods are well-described and easily replicable to a large extent. The results are accurately reported although figures need formatted/ changed in size. The discussion is well developed in the context of the limited literature in the area. Any of the queries I had were essentially answered in the limitations. As such I only have a few minor points for the authors and congratulate them on their manuscript
Title: I think the title should reference the population age e.g. …. In middle-older aged individuals
L11 ‘was’ a prospective
L125 Did the investigators ask if the participants were able to identify which type of mask was which before the study – i.e. did they know an N95 was an N95 mask etc.?
L162 Out of the 3 PFTs trials how was the data treated? – best effort; avg of all 3 etc.?
L220 remove (a) from Figure 1 as it doesn’t appear to refer to anything
Results: Not sure between the journal or the authors but the figures need to be made much larger as they’re incredibly difficult to read without zooming in multiple times. They also need to appear in-line with the text.
L333 Maybe worth mentioning the amount of indirect calculations made around VT, VO2 etc (L138-143), although the crossover design of the study protects this from having any real effect on outcomes likely.
Author Response
Thank you for your response and constructive feedback. Below are comments and edits made to the manuscript based on your review. Please let us know if there is anything else.
Title: I think the title should reference the population age e.g. …. In middle-older aged individuals
- See manuscript for update, changed to: "Effect of Face Masks on Physiological and Perceptual Responses During 30 Minutes of Self-Paced Exercise in Older Community Dwelling Adults"
L11 ‘was’ a prospective
- See manuscript for update, changed to: "This was a prospective experimental study that examined..."
In regards to: L125 Did the investigators ask if the participants were able to identify which type of mask was which before the study – i.e. did they know an N95 was an N95 mask etc.?
- Following a randomly assigned and counterbalanced fashion, the participants were informed on which mask they were wearing upon arrival to their scheduled exercise session.
L162 Out of the 3 PFTs trials how was the data treated? – best effort; avg of all 3 etc.?
- See manuscript for updated verbiage, we used participant's best effort for the analysis.
L220 remove (a) from Figure 1 as it doesn’t appear to refer to anything
- See manuscript for update, we removed "(a)" from the text.
Results: Not sure between the journal or the authors but the figures need to be made much larger as they’re incredibly difficult to read without zooming in multiple times. They also need to appear in-line with the text.
- Will need to consult with editor to see if we need to change the size. The figures were placed in manuscript based on template/outline given.
L333 Maybe worth mentioning the amount of indirect calculations made around VT, VO2 etc (L138-143), although the crossover design of the study protects this from having any real effect on outcomes likely.
- See manuscript for update, changed to: "Although the crossover design protects against any order of effects, the number of indirect calculations used to estimate VT and VO2 may produce values that are slightly different from the gold standard measurements."
Reviewer 2 Report
Firstly, thank you for the opportunity to review the manuscript. The work studies the effect of wearing face masks during exercise in older people. The introduction is well written and adequately supports the relevance of the study, especially for the population and context that we are living.
The methods are explicit and adequate and the results are well presented. The discussion and the interpretation of the results are well supported in the literature and virtually no remarks are to be made. In my opinion, the work is almost ready to be accepted.
I would only like to leave two remarks for consideration:
Average BMI values place women as overweight and men as obese, with an overall sample classification as overweight. To what extent this fact may influence how the results are interpreted?
The first paragraph of the discussion seems to be slightly contradictory (lines 258 – 272). On the one hand, you state that you found differences in HR between SM and NM, with no differences between the N95 and the NM condition. On the other hand, you state that the results are in line with studies that found no differences between wearing masks and not wearing them. Can you clarify this?
Congratulations on the work!
Author Response
Thank you for your response and constructive edits. Following are responses to your comments. Please let us know if there is anything else at this time.
Average BMI values place women as overweight and men as obese, with an overall sample classification as overweight. To what extent this fact may influence how the results are interpreted?
- See manuscript for updates. Updated limitations to include statement: "The results from our study may not be generalizable to older adults with a BMI less than 25 as the women from our study were classified as overweight while the men were classified as obese. "
The first paragraph of the discussion seems to be slightly contradictory (lines 258 – 272). On the one hand, you state that you found differences in HR between SM and NM, with no differences between the N95 and the NM condition. On the other hand, you state that the results are in line with studies that found no differences between wearing masks and not wearing them. Can you clarify this?
- See manuscript for updates. Clarified wording in discussion involving heart rate.
Reviewer 3 Report
General Comments
The aim of this study is to examine the effects of different types of masks on physiological and perceptual responses during 30-minutes of self-paced cycle ergometer exercise.
The research submitted is well designed, of great importance and innovative in its execution. However, when it comes to explaining what has been done, I have found certain shortcomings that should be improved to be published. These considerations are included in the specific comments.
I kindly ask the authors to read this report carefully and to respond accurately to the suggestions made if they consider them appropriate. I would like to thank you for your time and investment in this article and consider this review in the best possible way to improve and make science better.
Specific Comments
In the abstract results, there are references to different types of masks that have not been discussed previously in the method of the same abstract. This should be included.
The keywords must be different from those in the title.
Introduction
Between lines 36 to 42, I miss some references to support what is stated here. Not all countries proceeded in the same way.
Material and Methods
Participants are made aware of the inclusion criteria in the first paragraph of the method, but they are not explained until later. Either remove or explain the criteria
Reference is made to the two familiarisation sessions. What is the reason for the two? In the first section of the method, the second session is detailed but not the first.
Results
I suggest that table 1, being descriptive data on the sample, should be placed in the methods section.
Author Response
Thank you for your response and constructive feedback. Below are comments and edits made to the manuscript based on your review. Please let us know if there is anything else.
In the abstract results, there are references to different types of masks that have not been discussed previously in the method of the same abstract. This should be included.
- See manuscript for updates. Updated version reads: "different types of masks (no mask, surgical mask (SM), and N95-mask)"
The keywords must be different from those in the title.
- See manuscript for updates. Changed keywords to: "
Surgical Mask; N95-mask; Middle-Older Aged Individuals; Submaximal Exercise; Performance Measures"
Introduction
Between lines 36 to 42, I miss some references to support what is stated here. Not all countries proceeded in the same way.
- See manuscript for changes. Updated to: " Facemasks have traditionally been used in countries with high population densities and poor air quality, but they have become more important recently for use by some countries as a preventive measure against infectious diseases [3]. " Reference 3 is a metanalysis and systematic review that, according to the paper, "identified 172 observational studies across 16 countries and six continents, with no randomised controlled trials and 44 relevant comparative studies in health-care and non-health-care settings (n=25 697 patients)" in regards to transmission of viruses.
Material and Methods
Participants are made aware of the inclusion criteria in the first paragraph of the method, but they are not explained until later. Either remove or explain the criteria
- See manuscript for updates. Eliminated inclusion criteria text from line 100 to now read "Each participant attended..", as inclusion criteria clearly explained in 110-113.
Reference is made to the two familiarisation sessions. What is the reason for the two? In the first section of the method, the second session is detailed but not the first.
- See detailed description in manuscript of familiarization sessions in line 124-140. Please advise if you prefer different wording. Would be happy to revise with further clarification.
Results
I suggest that table 1, being descriptive data on the sample, should be placed in the methods section.
- See manuscript for update. Table moved to methods section.